# Convalescent plasma for treatment of COVID-19: study protocol for an open randomised controlled trial in Sweden

Joakim Dillner ![ORCID],[1,2] Johan Ursing[3]

## ABSTRACT

**Introduction** Although there are many studies on the use of convalescent plasma (CP) for treatment of COVID-19, it is not clear (1) which groups of patients may benefit, (2) what dose of plasma to give, or (3) which antibody levels the plasma should contain. Previous phase I/II studies and literature review suggest that CP should only be given to patients with viraemia, that a daily infusion should be given until the patient becomes virus free and that the neutralising antibody titre should preferably be >1:640

**Methods and analysis** An open randomised controlled trial enrolling patients with COVID-19, who must be SARS-CoV-2 positive in both airway and blood samples and admitted to a study hospital. Block randomisation 2:1 is to either 200 mL CP (preferably titre ≥1/640) daily for up to 10 days (until virus negative in blood) plus standard care or standard care only (control arm). The primary endpoint is mortality by day 28 after study inclusion. Secondary endpoints include mortality by day 60 and doses of plasma needed to clear viraemia. Assuming a reduced mortality of approximately 30% by the CP therapy and 85%–88% survival in the control arm, approximately 600 participants will be enrolled to the CP therapy arm and 300 participants to the control arm.

**Ethics and dissemination** Ethical approval has been granted by the Swedish Ethical Review Authority (reference: 2020-06277). Results from this trial will be compiled in a clinical study report, disseminated via journal articles and communicated to stakeholders.

**Trial registration number** NCT04649879.

¹Medical Diagnostics Karolinska, Karolinska University Hospital, Stockholm, Sweden
²Department of Laboratory Medicine, Karolinska Institutet, Stockholm, Sweden
³Department of Infectious Diseases, Danderyd University Hospital, Stockholm, Sweden

**Correspondence to**
Professor Joakim Dillner;
Joakim.Dillner@ki.se

## Strengths and limitations of this study

► A randomised trial design based on phase I and II studies that first investigated which patients to enrol (patients with viraemia) as well as how plasma should be characterised (high-neutralising antibody titre) and how it should be given (daily until viral clearance).

► Powered to detect a 30% mortality decrease.

► Clearance of viraemia is a secondary endpoint that will be very rapid to evaluate.

► Availability of high-titre convalescent plasma blood group matched to study participants may be a limiting factor.

► This is an unblinded trial.

Convalescent plasma (CP) therapy is a form of immunotherapy that has been used for the treatment and prevention of infectious diseases for more than 100 years. CP was successfully used in the treatment of SARS, Middle East respiratory syndrome and the 2019 H1N1 pandemic.[5–8] A meta-analysis of 32 SARS and severe influenza studies showed that CP treatment was associated with a significant reduction in the pooled odds of mortality (OR 0.25; 95% CI 0.14 to 0.45).[9] However, CP treatment did not significantly improve the survival of patients with Ebola disease.[10] This was possibly due to a lack of data on neutralising antibody titres.

Early studies on CP therapy suggested that in COVID-19, CP should be given daily until the patient becomes virus free.[11–14] In America, more than 70 000 patients have received CP. A safety report on 20 000 patients indicated that CP was well tolerated and not associated with severe adverse events (AE).[15] In a cohort of 1000 patients, mortality was significantly lower in patients receiving CP with antibody titres ≥1/640 compared with patients receiving CP with lower titres if the treatment was initiated within 72 hours of admission (data available at www.fda.gov). Though the numbers treated are large and

## INTRODUCTION

COVID-19 is an ongoing pandemic caused by the SARS-CoV-2. Current treatment options are limited. Corticosteroid therapy (dexamethasone, hydrocortisone or methylprednisolone) for up to 10 days has been shown to significantly reduce mortality in severely ill patients with COVID-19, summary OR 0.66 (95% CI 0.53 to 0.82).[1 2] Dexamethasone also reduced the risk of progression to invasive mechanical ventilation (relative risk 0.76 (0.61–0.96), p=0.021). However, no improvement was seen in patients not requiring supplemental oxygen therapy and corticosteroids were associated with prolonged viral shedding.[1 3 4]

the results suggest a beneficial effect, the lack of controls leaves space for much uncertainty. Moreover, there are randomised studies reporting no beneficial effect, although CP therapy was not specifically given to patients with viraemia in these trials.[16–18] There is thus a need for a randomised clinical trial that focuses on a population at high risk of severe COVID-19 (ie, patients with viraemia in early disease) and on which plasma to give and how (preferably neutralising antibodies of at least 1:640 and one plasma bag infused daily until clearance of viraemia occurs).

We found that viraemia on admission identifies patients at sevenfold increased risk of admission to intensive care and eightfold increased risk of death.[19] CP treatment appeared to result in rapid viral clearance in a small case series and two randomised trials.[16 18 20] CP appeared to be well tolerated in a phase I study in which patients only received one dose of CP and a phase II study in which CP was given until viraemia disappeared (J Ursing, unpublished data). Moreover, mortality was 0/6, 0/18 and 6/10 in patients who were not viraemic when treated, patients who were treated until viraemia disappeared and patients in whom viraemia continued despite treatment, respectively. The last group consisted of patients in whom treatment was stopped due to AEs, lack of CP or because they were part of the initial phase I study in which only one dose was given. Patients with viraemia may thus benefit from CP and can be identified early prior to development of severe COVID-19.

## Rationale

CP has been shown to be safe and effective for treatment of several diseases. Data indicate that it is safe for treatment of COVID-19. Preliminary data indicate that treatment should be given early prior to development of severe illness. A group at high risk of severe disease and death that has the most to benefit from CP can be easily identified by detection of viraemia on admission.[19] Phase II study data indicate that treatment should be given until SARS-CoV-2 is no longer detected in serum and the donor antibody neutralisation titres should be ≥1/640. A randomised controlled trial in which patients with viraemia are treated with CP with the equivalent of an antibody titre ≥1/640 is thus required to determine if CP can be an effective COVID-19 treatment.

## Objective

The aim is to assess the efficacy, safety and tolerability of CP for treatment of patients with COVID-19 and viraemia.

## METHODS AND ANALYSIS
### Study design

A multicentre, open 2:1 randomised controlled clinical trial conducted in Sweden. Study sites include Karolinska University Hospital, Danderyd Hospital and the Hospital in Falun (for a complete list of study sites, visit Clinical-Trials.gov). All Hospitals with infectious diseases departments in Sweden have been invited to participate. The study will be conducted in accordance with the Helsinki Declaration on ethical principles for medical research involving human subjects. The study is expected to take place during 2021 (3 December 2020 to 1 February 2022). This study protocol is prepared using the Standard Protocol Items: Recommendations for Interventional Trials guidelines.[21]

### Participants

Patients with COVID-19 admitted to a study hospital will be considered for study entry. Patients will be identified by notification among doctors at the hospitals. Patients will be informed of the study and offered to participate if they match the inclusion and exclusion criteria. An electronic (REDCap, a web-based service for electronic data collection) case record form will be filled in for all screened and included patients. On inclusion, an informed consent form will be signed. Enrolment of patients is done by dedicated physicians and expected to take place during a 12-month period. Intervention is taking place up to a maximum of 10 consecutive days and follow-up will be on day 28 and day 60 after inclusion (table 1). Adherence to study protocol will be monitored by registry linkages with the patient registry.

### Eligibility criteria

Inclusion criteria for participants:
- ► Age ≥18.
- ► Admitted to a study hospital.
- ► Active COVID-19 defined as symptoms+SARS-CoV-2 identified from upper or lower airway samples and blood.
- ► Negative pregnancy test taken before inclusion and use of an acceptable effective method of contraception

**Table 1** Planned questionnaire and sampling follow-up

| Time point (days) | Prior to or on inclusion | Daily during treatment | 28 | 60 |
|---|---|---|---|---|
| Informed consent | X | | | |
| CRF (Case Record Form) | X | X | X | X |
| Routine blood chemistry | X | | | |
| Serum sample | X | X | | |
| Virus PCR in blood | X | X | | |

until treatment discontinuation if the participant is a woman of childbearing potential.

► Written informed consent after meeting with a study physician and ability and willingness to complete follow-up.

Exclusion criteria for participants:

► No matching plasma donor (exact matching in the ABO blood group system is required).
► Unavailability of plasma.
► Estimated glomerular filtration rate <30 (kidney failure stage III or more).
► Pregnancy (urinary hcg (human chorionic gonadotropin)).
► Breast feeding.
► Inability to give informed consent.

The study will be conducted at wards in study hospitals where patients with COVID-19 are treated. To be eligible to participate, the personnel involved in the study will receive appropriate training relevant to the study.

## Intervention

Patients will be randomised 2:1 to treatment with CP plus standard care or standard care only. Randomisation is by random permutated blocks using REDCap or equivalent.

Participants will receive 200 mL CP daily until SARS-CoV-2 is no longer detectable in the blood up to a maximum of 10 CP infusions. CP will be given as a slow infusion over 2 hours. CP neutralisation titre of ≥1/640 (which is higher than conventionally used as high-titre plasma) or an ELISA reactivity against the Spike protein of SARS-CoV-2 by the Euroimmun commercial assay >9 is desired. The in-house neutralisation test is performed as described.[22]

New antibody tests are under development and can be used instead if equivalence to neutralisation or Euroimmun ELISA is demonstrated. If no high-titre plasma is available, lower antibody titre plasma may be given but must in no instance be lower than 4 in the Euroimmun ELISA. CP will be obtained from respective study sites blood bank. If this is not possible, CP will be available from the Department of Clinical Immunology and Transfusion Medicine at Karolinska University Hospital.

Patients will be monitored for AEs, especially allergic reactions and impaired renal function. Patients not able to receive study treatment will receive standard care.

If an attending physician assesses that a patient is suffering from an AE induced by the plasma infusion and requiring treatment, the infusion will be stopped for that patient. The outcome will be classified as an AE necessitating treatment stop.

A study subject can terminate his/her participation in the study at any time without giving a reason why. Should a subject wish to do so, he/she will be treated in line with standard recommendations.

Study organisers can end a patient's participation for safety reasons.

## Outcomes

Primary endpoint: mortality by day 28 after inclusion into the study.

Secondary endpoints:

► Mortality by day 60 after inclusion into the study.
► Requirement of invasive ventilation or $PaO_2/FiO_2$ ≤70 for ≥12 hours in the case of patients not eligible for intensive care.
► AEs.
► Dose of plasma needed to clear viraemia.
► Time to clearance of viraemia.

## Sample size

Available data indicate that CP is safe and that high-titre CP can reduce mortality by approximately 30%. For that reason, a 2:1 randomisation is preferred. Randomisation will be done by computer-generated random numbers. Based on the available data, we assume 85%–88% survival in the control arm. Assuming a power of 80% and an alpha of 0.05%, 920 patients need to be enrolled.

## Recruitment

To reach the number of patients needed to conduct the study with enough power, all hospitals in Sweden with patients acutely ill in COVID-19 are invited to participate.

## Measurements

Blood samples for analysis of the following routine blood chemistry will be taken prior to start of treatment if available at respective study sites: C-reactive protein, procalcitonin, white cell count, haemoglobin, creatinine, creatine kinase, D-dimer, troponin, Alanine aminotransferase (ALT), Aspartate aminotransferase (AST), Lactate dehydrogenase (LD) and bilirubin.

Blood samples for detection of SARS-CoV-2 in the blood will be taken prior to treatment start, daily during treatment and until two consecutive negative results are obtained. Serological samples will be collected for subsequent SARS-CoV-2 culture whenever PCR samples are taken.

## Data collection

The following basic data will be collected and entered into electronic records (REDCap):

► Age, gender and reason for non-inclusion for screened but not included patients.
► Date of symptom onset and date of start of CP therapy will be recorded and used in subgroup analyses.
► Basic patient characteristics, comorbidities, disease severity characteristics and treatment on study entry.
► Each dose of CP and date it was given along with any observed AEs.
► Concomitant treatment with antivirals and/or corticosteroids.
► The outcome of daily SARS-CoV-2 PCR until two consecutive negative PCRs are recorded.
► The need for mechanical ventilation and date when this was initiated.
► For patients not eligible for intensive care: each day when $PaO_2/FiO_2$ ratio was less than 70 for ≥12 hours.

$PaO_2$ and $FiO_2$ will be estimated from $SO_2\%$ and $O_2$ flow in nasal cannula, face mask or face mask with reservoir based on medical record data. A ratio of 70 is approximately equal to 90% $SO_2$ with 8–9 L of oxygen flow using a face mask with a reservoir.

► Date of death.
► Survival or death by day 28 and day 60.

Data are entered directly into a computerised system that includes automatic range checks.

### Data analysis plan

Efficacy outcomes will be determined using Cox regressions analyses, non-parametric regression analyses and survival analyses of the cohort of patients that fulfilled entry criteria but not exclusion criteria. A per-protocol and intention-to-treat analysis will be done. Final analysis will include a description of included participants, proportions of AEs and any serious AEs, the proportion of participants withdrawn or lost to follow-up, the cumulative success and failure rates by day 28 and 60. Categorical variables will be compared using the $\chi^2$ test or Fisher's exact test and continuous variables using quantile regression. The final data analysis will be performed when a total number of 88 outcome events have occurred. An interim analysis will be performed after 44 outcome events.

### Patient and public involvement

No patient or patient organisation was involved in design, recruitment or management of the study.

### ETHICS AND DISSEMINATION

The trial has been registered with ClinicalTrials.gov, and sponsor protocol number CP03. The investigational product is a blood product (not classified as a drug) and the regulatory agency is the National Inspection Authority for Health and Care, which has approved the trial. The trial has been approved by the National Ethical Review Agency (decision 2020-06277). Patients will be informed of the study verbally and in writing. Details about the trial including risks and benefits will be explained and any questions answered, including the information that all participants in the study are covered by the regular patient injury insurance. A copy of the written consent form will then be signed. If an amendment substantially alters the study design, or increases the potential risk to the patients, written informed consent must be obtained again for currently enrolled patients and must be provided to additional patients prior to their entry into the study.

The trial will be conducted in compliance with the ICH (The International Council for Harmonisation of technical requirements for pharmaceuticals for human use) guidelines for Good Clinical Practice, the latest edition of the Declaration of Helsinki and any local regulations.

On inclusion, an entry will be made in the patient's medical record specifying that the subject is participating in this study, if CP treatment is given and that written informed consent has been obtained. Any apparent side effects experienced by the subject will be assessed from the time the subject signs the informed consent and as long as the subject is part of the study, and will be reported by study site personnel either as a baseline event or an AE. The study personnel will document any baseline events or AEs in the electronic case record form (eCRF), whether observed by the investigator or reported by the subject. The safety of the different treatment arms will be assessed with regard to AEs, baseline medical conditions and findings from the physical examination and laboratory tests. All AEs, irrespective of nature, will be followed until resolution or until end of the follow-up period.

The reporting period for AEs starts at inclusion and ends at the final follow-up visit 2 months after inclusion. Interim analyses will be done after inclusion of 100, 300 and 600 patients. The study will be stopped if a 30% difference in treatment outcome is detected and an independent data monitoring committee will be convened to review the data and determine if the study should be stopped or not. An external party will monitor the study regularly and ensure that the study follows Good Clinical Practice. Audits or inspections will be performed at the study site during or after the study. These visits may include source data verification and confidentiality documents are therefore created.

After completion of the study, a clinical study report will be prepared that will form the basis for a manuscript intended for publication in a medical journal. Data will only be reported in aggregated form, without being able to be attributed to an individual. Records of the study will be kept for 10 years after final signing of the clinical study report. Data will be entered into a database by the investigators at the coordinating hospital (Danderyd Hospital). Study ID will only be used on computerised records. The key will be stored with the case record forms (CRFs) at Danderyd Hospital. Only the sponsor, principal investigator and participating researchers will have access to all data. Source data comprising electronic medical records will be stored at each study site.

**Contributors** JD and JU conceived the study and wrote the paper. Both authors approved the final version of the manuscript.

**Funding** This work was supported by the Swedish Research Council (2020-05793_3).

**Disclaimer** The funding agency has had no role in the design, execution or interpretation of the study or in the decision to submit for publication.

**Competing interests** None declared.

**Patient and public involvement** Patients and/or the public were not involved in the design, or conduct, or reporting, or dissemination plans of this research.

**Patient consent for publication** Not required.

**Provenance and peer review** Not commissioned; externally peer reviewed.

**ORCID iD**
Joakim Dillner http://orcid.org/0000-0001-8588-6506

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
