## [Reviewer comments · BMJ Open]

ARTICLE DETAILS

TITLE (PROVISIONAL)	Convalescent plasma for treatment of COVID-19: Study protocol for an open randomised controlled trial in Sweden
AUTHORS	Dillner, Joakim; Ursing, Johan

VERSION 1 – REVIEW

REVIEWER	Joyner, Michael Mayo Clinic, Department of Anesthesiology
REVIEW RETURNED	17-Feb-2021

GENERAL COMMENTS	This is a clearly conceived and well written protocol for aggressive convalescent plasma therapy in COVID-19 patients. I have 2 major caveats for the authors: 1) Emerging data for many forms of antibody therapy including CP indicates "early is better", and at this time in many countries patients admitted to hospital have already been sick for sometime. How will duration of disease be considered and controlled for?2) It was unclear to me if the endogenous antibody response prior to treatment will be measured (clearly the needed blood samples will be collected). This is important because beyond duration of disease, the presence of endogenous antibodies may also be a key factor in the efficacy of CP and other forms of antibody therapy. This is highlighted by the fact that patients who have inborn or acquired diseases that limit their ability to generate endogenous antibodies seem to do very well when treated with CP.
---

REVIEWER	Franchini, Massimo Carlo Poma Hosp
REVIEW RETURNED	19-Feb-2021

GENERAL COMMENTS	I found the protocol potentially interesting. I suggest to the authors to further improve their protocol by adding this information: 1) Intervention: The authors defined high titer CP as > 640. However, high titre CP is conventionally defined as >160. The authors should clarify this issue.2) Intervention. The authors should describe in details the neutralization assay used.3) Intervention. The authors permit the use of lower titre CP. However, they should defined the minimum titer tolerated.4) Outcomes. To assess the outcomes correctly, it is important to know the degree of severity of COVID-19 and the time of CP infusion from symptom onset/hospitalization. It is indeed recognized that high titre CP works better if administered early
---

	during COVID-19. All this information is essential to perform subgroup analyses.
--	--

VERSION 1 – AUTHOR RESPONSE

Thank you for your decision that you are willing to consider a revised manuscript that takes the reviewers' comments into account. We have revised the manuscript as detailed point by point below and hope that the manuscript will now meet with your acceptance.

Reviewer #1: Michael Joyner, Mayo Clinic

Comments to the Author:

This is a clearly conceived and well written protocol for aggressive convalescent plasma therapy in COVID-19 patients. I have 2 major caveats for the authors:

1) Emerging data for many forms of antibody therapy including CP indicates "early is better", and at this time in many countries patients admitted to hospital have already been sick for sometime. How will duration of disease be considered and controlled for?

***We have added that the number of days between symptom onset and start of CP therapy will be recorded and used in subgroup analyses.

2) It was unclear to me if the endogenous antibody response prior to treatment will be measured (clearly the needed blood samples will be collected). This is important because beyond duration of disease, the presence of endogenous antibodies may also be a key factor in the efficacy of CP and other forms of antibody therapy. This is highlighted by the fact that patients who have inborn or acquired diseases that limit their ability to generate endogenous antibodies seem to do very well when treated with CP.

***This will not be measured. This was a deliberate decision as it was felt that any delays in start of therapy could be deleterious.

Reviewer #2: Dr. Massimo Franchini, Carlo Poma Hosp

Comments to the Author:

I found the protocol potentially interesting.

I suggest to the authors to further improve their protocol by adding this information:

1) Intervention: The authors defined high titer CP as > 640. However, high titre CP is conventionally defined as >160. The authors should clarify this issue.

***We have now stated that we use a higher titer than conventionally.

2) Intervention. The authors should describe in details the neutralization assay used.

***This is now referenced.

3) Intervention. The authors permit the use of lower titre CP. However, they should defined the minimum titer tolerated.

***This is now given.

4) Outcomes. To assess the outcomes correctly, it is important to know the degree of severity of COVID-19 and the time of CP infusion from symptom onset/hospitalization. It is indeed recognized that high titre CP works better if administered early during COVID-19. All this information is essential to perform subgroup analyses.

*** We have added that the number of days between symptom onset and start of CP therapy will be recorded and used in subgroup analyses.